# Switching from Electron to Hole Transport in Solution-Processed Organic Blend Field-Effect Transistors

**DOI:** 10.3390/polym12112662

**Published:** 2020-11-11

**Authors:** Julia Fidyk, Witold Waliszewski, Piotr Sleczkowski, Adam Kiersnowski, Wojciech Pisula, Tomasz Marszalek

**Affiliations:** 1Department of Molecular Physics, Faculty of Chemistry, Lodz University of Technology, Zeromskiego 116, 90-924 Lodz, Poland; julia.fidyk@dokt.p.lodz.pl (J.F.); witold.waliszewski@dokt.p.lodz.pl (W.W.); piotr.sleczkowski@p.lodz.pl (P.S.); pisula@mpip-mainz.mpg.de (W.P.); 2Leibniz Institute of Polymer Research, Hohe Str. 6, 01069 Dresden, Germany; kiersnowski@ipfdd.de; 3Wroclaw University of Science and Technology, Wybrzeze Wyspianskiego 27, 50-370 Wroclaw, Poland; 4Max Planck Institute for Polymer Research, Ackermannweg 10, 55128 Mainz, Germany

**Keywords:** organic electronics, bulk heterojunction, charge carrier transport, organic field-effect transistor, film morphology

## Abstract

Organic electronics became an attractive alternative for practical applications in complementary logic circuits due to the unique features of organic semiconductors such as solution processability and ease of large-area manufacturing. Bulk heterojunctions (BHJ), consisting of a blend of two organic semiconductors of different electronic affinities, allow fabrication of a broad range of devices such as light-emitting transistors, light-emitting diodes, photovoltaics, photodetectors, ambipolar transistors and sensors. In this work, the charge carrier transport of BHJ films in field-effect transistors is switched from electron to hole domination upon processing and post-treatment. Low molecular weight n-type N,N′-bis(n-octyl)-(1,7&1,6)-dicyanoperylene-3,4:9,10-bis(dicarboximide) (PDI8-CN_2_) was blended with p-type poly[2,5-bis(3-tetradecylthiophene-2-yl)thieno[3,2-b]thiophene] (PBTTT-C_14_) and deposited by spin-coating to form BHJ films. Systematic investigation of the role of rotation speed, solution temperature, and thermal annealing on thin film morphology was performed using atomic force microscopy, scanning electron microscopy, and grazing incidence wide-angle X-ray scattering. It has been determined that upon thermal annealing the BHJ morphology is modified from small interconnected PDI8-CN_2_ crystals uniformly distributed in the polymer fraction to large planar PDI8-CN2 crystal domains on top of the blend film, leading to the switch from electron to hole transport in field-effect transistors.

## 1. Introduction

Electronics based on organic semiconductors (OSCs) offer advantages such as mechanical flexibility, low cost and large-area fabrication by solution processing [1,2,3,4,5]. Organic light-emitting diodes, solar cells, memory devices and organic field-effect transistors (OFETs) are constantly improved via synthesis of new semiconductors as well as development of novel device architectures, thin-film processing and postprocessing techniques [5,6,7,8,9,10]. OFETs exhibiting charge carrier mobilities comparable to amorphous silicon (>1 cm^2^/Vs) became attractive for future applications in electronics, circumventing conventional vacuum and photolithographic fabrication processes. Especially unipolar [11,12] and ambipolar [13,14,15,16] OFETs became important for complementary logic circuits and inverters. Although most OSCs intrinsically conduct electrons and holes, the vulnerability of electron carriers to ambient conditions (i.e., the detrimental effects of O_2_ and H_2_O) results in serious degradation of the n-type transport in devices [4,17]. Additionally, an alignment of the electrode work function with the HOMO and LUMO levels of the semiconductor is necessary for an efficient injection of electrons and holes into the active layer [18,19,20]. This means that the type of unipolar conduction, n- or p-type, of an organic semiconducting film is fixed after the device fabrication and a subsequent modification is challenging due to the misalignment of the metal work function. 

An alternative approach consists of blending two organic semiconductors into a bulk heterojunction (BHJ) morphology with phase separated p- and n-type fractions [16,21,22]. This method is used frequently not only in light emitting transistors, light-emitting diodes and photovoltaics [22,23,24,25,26,27], but also applied to achieve a balanced transport of electrons and holes in ambipolar OFETs [28]. In most cases, BHJs are fabricated from a solution of polymer−polymer or polymer−low molecular weight semiconductor blend. The BHJ morphology strongly determines the charge carrier transport in OFETs. Since the charge carrier transport in OFETs occurs parallel to the substrate, the BHJ interface needs to be adapted accordingly to avoid scattering of charges and interruption in the charge carrier path [28]. Further essential requirements are a contact of each semiconductor phase with the source and drain electrodes and a continuous percolation path for each type of charge carrier. Most studies focus on the impact of the blend ratio of the semiconducting components on the BHJ film morphology and charge carrier transport in OFETs [28,29,30]. However, the influence of processing parameters has not been studied well so far.

So far, BHJs have only been used to fabricate ambipolar OFETs [16,28,31] with a balanced transport of holes and electrons. In our novel approach, BHJs are exploited to control the type of unipolar conduction by tuning the blend morphology, even after device fabrication. Moreover, we demonstrate for the first time the possibility of switching from unipolar electron to unipolar hole transport in the solution-processed BHJ OFETs. The transition from n- to p-type device behavior is achieved by well-controlled changes in the BHJ morphology induced through thermal post-treatment. The possibility of switching the conduction type in solution-processed OFETs by post-treatment [32] is important for the fabrication of logic circuits such as complementary inverters. In this work, we demonstrate the possibility of switching from electron to hole transport in solution-processed BHJ OFETs. The transition from n- to p-type device behavior is achieved by controlled changes in the BHJ morphology induced through thermal post-treatment. Blends of p-type poly[2,5-bis(3-tetradecylthiophene-2-yl)thieno[3,2-b]thiophene] (PBTTT-C_14_) and n-type low molecular weight N,N′-bis(n-octyl)-(1,7&1,6)-dicyanoperylene-3,4:9,10-bis(dicarboximide) (PDI8-CN_2_) were solution cast to form PBTTT-C_14_:PDI8-CN_2_ BHJ films in OFETs. Both compounds were chosen due to their chemical stability, good field-effect mobility, and solution processability [33,34,35,36,37,38]. Systematic variation of rotation speed, solution temperature, and thermal annealing allowed the identification of key processing parameters to control the BHJ film morphology. It was observed that a prolonged drying time and lower spin-coating speed facilitated the formation of larger PDI8-CN_2_ crystals, while a high solution temperature led to the growth only of small PDI8-CN_2_ crystals immersed in the polymer fraction. Thermal annealing after the film deposition significantly changed the blend morphology and induced a switch from electron to hole transport in OFETs.

## 2. Materials and Methods

PBTTT-C_14_ was purchased from Sigma-Aldrich (Saint Louis, MO, USA) and PDI8-CN_2_ as ActivInk N1200 from Polyera Corporation (Skokie, IL, USA) (Figure 1a,b). Both compounds were used as received. The HOMO level of −5.1 eV for PBTTT-C_14_ [39] and LUMO of −4.3 eV for PDI8-CN_2_ [40] were properly aligned with the work function of gold (WF = −5.1 eV) for injection of electrons and holes into the BHJ film (Figure 1c). The energy gap between these levels was less than 1 eV preventing formation of the charge carrier traps at semiconductor/electrode interface [16,20,41]. It has been reported that the effective energy gap allowing charge transfer and increase in the conductivity is below 0.6 eV [42,43]. Due to the relatively high energy gap of 0.8 eV between HOMO of PBTTT-C_14_ (−5.1 eV) and LUMO of PDIC8-CN_2_ (−4.3 eV) a charge transfer was excluded in our case. A concentration of 20 mg/mL in 1,2-dichlorobenzene of PBTTT-C_14_ and PDI8-CN_2_ (50/50 wt %) was used for the solution processing.

Highly p-doped silicon wafers with thermally grown 300 nm silicon dioxide (capacitance of 11 nF cm^−2^) were used as substrates for the device fabrication. The substrates were ultrasonically cleaned using standard procedure (15 min acetone, 15 min isopropanol) and dried with N_2_. BHJ films were deposited by spin-coating at different rotation speeds ranging from 1000 to 6000 rpm at a solution temperature from 80 to 180 °C in the glovebox in nitrogen atmosphere. Samples fabricated at different solution temperatures were annealed on a hotplate in the glovebox for 1 h at 140 °C. For the investigation of the post-treatment on the BHJ morphology, samples fabricated at 3000 and 4000 rpm from solution at 100 and 120 °C were annealed at 200 °C for 2 h on a hotplate in the glovebox. For one set of samples fabricated at 4000 rpm from solution at 100 °C, the substrates were additionally modified by octadecyltrichlorosilane (OTS) (Sigma Aldrich Chemie Gmbh, Munich, Germany). After standard cleaning, these substrates were immersed in piranha solution (a mixture of 3:7 (v/v) of 30% H_2_O_2_ and 98% H_2_SO_4_) for 30 min. Subsequently, the silicon wafers were rinsed a few times with deionized water and methanol, dried with the nitrogen flow and immersed in an unstirred solution of 0.5 % OTS in a mixture of chloroform and hexane (1:4 (v/v)) for 30 min. After the modification, the substrates were additionally rinsed with chloroform and dried with nitrogen flow.

Bottom gate top contact (BGTC) configuration was employed for the BHJ OFETs. Source-drain gold electrodes were thermally evaporated through shadow masks to the nominal thickness of 80 nm (controlled using quartz crystal microbalance). OFETs with channel width (W) of 1 mm and variable channel lengths (L) of 10–30 µm were fabricated and further characterized using Keithley 4200 source meter (Keithley Instruments, Inc., Cleveland, OH, USA) connected to a needle-probe station. Charge carrier mobilities were calculated from the transfer characteristics in the saturation regime, using the following formula [44]:$$\mu_{sat}=\frac{2I_{DS}L}{WC_{i}{\left(V_{G}-V_{th}\right)}^{2}}$$
where *μ_sat_* stands for the charge carrier mobility, *I_DS_* is the drain current, *W* the channel width, *L* the channel length, *C_i_* the capacitance of gate dielectric, *V_G_* the gate voltage and *V_th_* the threshold voltage.

The LEO Gemini 1530 Scanning Electron Microscope (SEM) (Carl Zeiss AG, Oberkochen, Deutschland) and Veeco Dimension 3100 Atomic Force Microscope (AFM) (Digital Instruments, Santa Barbara, CA, USA) were used to investigate the morphology and thickness of the BHJ thin films. Grazing incidence wide-angle X-ray scattering (GIWAXS) was performed at the DELTA Synchrotron (Dortmund, Germany), beamline BL09. The photon energy was set to 10 keV (λ = 1.24 Å). During the measurements, the samples were under vacuum (~1 mbar). The incident angle (α_i_) of the X-ray beam was adjusted individually for each sample in the range of 0.19°–0.20°. The scattered intensity was recorded by a 2D image plate (MAR345, marXperts GmbH, Norderstedt, Germany). The exposure time was 800 s. The *q*-range (*q* = 4 × π × sin θ × λ^−1^) was calibrated using silver behenate standard. The data was processed and analyzed using Datasqueeze (University of Pennsylvania, Philadelphia, PA, USA) and OriginPro (OriginLab Corporation, Northampton, MA, USA).

## 3. Results and Discussion

### 3.1. Role of Rotation Speed

The molecular organization and the crystallization kinetics of the conjugated molecules can be controlled during spin-coating parameters through tuning the amount of residual solvent after the film formation. Since the amount of solvent and drying time are closely related to the spin-coating speed, in this study, the BHJ films were deposited at various rotation speeds between 1000–6000 rpm. A solution temperature of 100 °C ensured sufficient dissolution and mixing of the semiconductors. Scanning electron microscopy (SEM) images in Figure 2 show the effect of the rotation speed on the crystallization of PDI8-CN_2_ at the top film surface in the presence of PBTTT-C_14_. Although the rotation speed did not affect the PBTTT-C_14_ morphology, a higher rotation speed reduced the amount of solution remaining on the substrate during spin-coating and decreased the drying time leading to the growth of only small PDI8-CN_2_ crystals. Additionally, less solvent remaining in the blend lowered the recrystallization process.

As evident from Figure 2, in nonannealed BHJ films the PDI8-CN_2_ crystal size decreased with increasing rotation speed. Concerning the BHJ formation, the interconnected and uniformly distributed PDI8-CN_2_ crystals in the PBTTT-C_14_ fraction ensured an electron transport in the OFETs [28]. Typical n-type transfer characteristics are presented in Appendix A. Despite evident differences in BHJ morphology by varying the rotation speed, only minor changes in field-effect mobility were observed (Appendix A). The highest n-type mobility of 2 × 10^−4^ cm^2^/(Vs) was determined for rotation speeds of 3000 and 4000 rpm. In these conditions, the optimum size and distribution of the PDI8-CN_2_ crystals created the necessary pathways for electrons. However, no hole transport was observed for this series of transistors.

### 3.2. Impact of Solution Temperature

The solution is usually heated to improve the solubility of semiconductors and to prevent aggregation prior to deposition [34] or to accelerate the solvent evaporation. In this study, the effect of solution temperature, which varied from 80 to 180 °C, on the BHJ morphology was investigated for the rotation speed of 3000 rpm leading to the highest n-type mobility within the series. Figure 3 shows atomic force microscopy (AFM) images of the surface of BHJ films obtained for different solution temperatures at 3000 rpm. After the deposition, each sample was thermally annealed at 140 °C for 1 h to completely remove solvent residues.

The increase in solution temperature from 80 to 140 °C and subsequent annealing did not affect the surface morphology of the PBTTT-C_14_:PDI8-CN_2_ BHJ. However, for films cast at 160 and 200 °C, the deposition temperature was higher than the annealing temperature. For this reason, PDI8-CN_2_ could not recrystallize during the post-treatment. Additionally, the increase of the solution temperature accelerated the solvent evaporation during the deposition. For temperatures above 140 °C, the growth kinetics of PDI8-CN_2_ were much faster leading to significantly smaller crystal needles immersed in the PBTTT-C_14_ phase.

Despite the interconnected PDI8-CN_2_ crystals in the annealed BHJ films for solution temperatures below 160 °C, only a hole transport was found (Appendix A and Appendix A). Although the surface morphology of the BHJs did not differ significantly in the temperature range of 80–140 °C, a maximum hole charge carrier mobility of 1 × 10^−3^ cm^2^/(Vs) was determined for 100 °C (Figure 4). For samples cast above 140 °C no field effect was observed. Evident changes in blend morphology were mainly ascribed to the size of PDI8-CN_2_ crystals, while the PBTTT-C_14_ morphology remained unaffected. It was assumed that despite similarities in the film topography, the solution temperature influenced the morphology and molecular organization of the bulk BHJ film. Since the charge carrier transport in OFETs occurs near the dielectric/semiconductor interface [45], differences in the interface morphology can lead to changes in the field-effect mobility.

### 3.3. Effect of Thermal Annealing

The annealing process is mostly used to remove residual solvent and to improve the molecular organization in thin organic semiconducting films before their use in electronic devices [16,46]. For a better understanding of the change from electron to hole transport in the BHJ films observed after thermal post-treatment, the annealing temperature was increased to 200 °C (Appendix A) above the PBTTT-C_14_ transition to the liquid crystal phase [35,36,37]. For an annealing temperature of 140 °C, which is lower than the solution temperatures of 160 and 200 °C, noticeable differences in the PBTTT-C_14_:PDI8-CN_2_ morphology were observed (Figure 3). However, if the annealing temperature of 200 °C was higher than the solution temperature, the film morphology of the BHJ was independent of the lower solution temperature (Appendix A). Although the morphology was similar to blend films annealed at 140 °C, the elevated temperature of 200 °C enabled recrystallization of PDI8-CN_2_ and also in blends cast at 160 and 200 °C. The PBTTT-C_14_ polymer backbones became more mobile at film annealing at 200 °C above the liquid crystal phase transition, facilitating a reorganization of PDI8-CN_2_ into needle-like crystals on the surface of the BHJ film.

The morphology transition of the optimized-BHJ films (fabricated at 3000 rpm from solutions at 100 or 120 °C) upon annealing is evident from SEM images in Figure 5. Before annealing, the blends revealed a network of interconnected PDI8-CN_2_ crystals distributed in the PBTTT-C_14_ fraction (Figure 2 and Figure 5a,c). Annealing at 200 °C promoted growth of the PDI8-CN_2_ crystals and led to evident phase-separation of blend components as marked by arrows in Figure 5d. The recrystallization of PDI8-CN_2_ upon annealing resulted in larger crystals (Figure 5b,d) as confirmed by GIWAXS results (Figure 6).

The SEM images (Figure 5a,c) show small PDI8-CN_2_ crystals in the as-cast films as confirmed by the low intensity of corresponding GIWAXS peaks and their relatively broad azimuthal distribution as well as Scherrer’s widths (Figure 6). Thermal annealing of the films at 200 °C initiated a crystal growth which was manifested by a significant increase of intensity as well as narrowing of the azimuthal distribution and Scherrer breadths of the PDI8-CN_2_ peaks visible in the GIWAXS patterns (Figure 6). A similar effect was observed in P3HT-PDI8-CN_2_ blends where an Ostwald ripening-like aggregation of smaller crystals forming eventually large crystal domains was reported [47]. Heating the film above the liquid crystal phase transition of PBTTT-C_14_ reduced the viscosity of the film, which probably facilitated migration of PDI8-CN_2_ to its surface and enabled effective crystal growth on top. As revealed by both SEM and GIWAXS (Figure 5 and Figure 6) a crystal growth through Ostwald ripening enhanced the planar growth of PDI8-CN_2_ crystals on top of the BHJ films. In addition to changes in the PDI8-CN_2_ morphology, annealing of the films influenced the crystallinity of the polymer. The GIWAXS patterns in Figure 6 reveal an ordered structure and edge-on dominated chain orientation of PBTTT-C_14_ both in pure polymer and in the PBTTT-C_14_:PDI8-CN_2_ films. The positions of the meridional (h00) Bragg series (Figure 6a) and the equatorial diffuse reflection profile corresponding to the π-stacking distance (d_π-__π_) suggested that alkyl side chains of PBTTT-C_14_ were (at least partly) twisted causing a large separation between the backbones [46]. Deconvolution of the diffuse reflection in the range 1.3–1.7 Å^−1^ revealed the existence of two components: d_π-__π1_ = 4.4 Å and d_π-__π2_ = 3.9 Å, which corresponded to twisted and non-twisted side chain configurations, respectively, in the PBTTT-C_14_ domains [48]. Annealing of the PBTTT-C_14_ film did not cause significant changes in the polymer packing but caused the polymer crystal domains to grow, mainly along the *a*-lattice vector. That growth was manifested by narrowing of the out-of-plane (meridional) diffraction peaks (cf. Figure 6a,b). Addition of PDI8-CN_2_ and formation of the BHJ films did not change the crystal packing of the polymer, but significantly reduced the PBTTT-C_14_ average crystal domain size in the as-cast films. This decrease can be concluded from two facts: (i) the meridional peaks for PBTTT-C_14_ were significantly broader in the case of BHJ film than for the pure polymer (Figure 6a,c), and (ii) the higher order (300 and 400) peaks were missing in the pattern of the as-cast BHJ film (Figure 6c). The strong amorphous halo visible in Figure 6c suggested that in the as-cast BHJ films a large fraction of PBTTT-C_14_ remained disordered. Annealing of the BHJ films caused crystals to grow. Comparison of Figure 6b,d indicated that the h00 (meridional) peaks observed for the annealed pure PBTTT-C_14_ film (Figure 6b) were broader than those for the annealed BHJ film (Figure 6d). This suggested that the size of polymer crystal domains in the annealed BHJ film was larger than in the pure PBTTT-C_14_ film.

The electrical evaluation of the BHJ films before and after thermal annealing at 200 °C confirmed the change of conductivity from electron to hole transport. Figure 7 shows output characteristics for the PBTTT-C_14_:PDI8-CN_2_ blend film before and after annealing at 200 °C (transfer characteristics are presented in Appendix A). The strongest difference in electrical properties before and after annealing was observed for the rotation speed of 4000 rpm and solution temperature of 100 °C (Figure 7 and Appendix A). Before annealing, electron transport was dominating, while after annealing only hole transport was observed.

In the BHJ films two possible conductive paths were formed: for holes through the PBTTT-C_14_ phase and for electrons via the network of PDI8-CN_2_ crystals. Before annealing, dominant electron transport was established by the PDI8-CN_2_ crystals, while at the same time the hole conduction was limited due to disordered PBTTT-C_14_. When the BHJ film was annealed at 200 °C in the liquid crystalline phase of PBTTT-C_14_, the order of the polymer chains was improved, especially in the π-stacking direction [36,37]. The phase separation between PBTTT-C_14_ and PDI8-CN_2_ was also improved leading to the formation of larger PDI8-CN_2_ crystals which were visible in the SEM images (Figure 5b,d). On the other hand, PDI8-CN_2_ recrystallized into crystal sizes below 25 μm which was smaller than the applied channel length of the OFETs. Because the PDI8-CN_2_ crystals were too small to connect the source and drain electrodes, electron transport was not observed. Before annealing, an electron charge carrier mobility of 1 × 10^−4^ cm^2^/(Vs) was determined, while after annealing a hole mobility of 2 × 10^−3^ cm^2^/(Vs) was found. Based on the S-shape of the output characteristics in the positive range of the applied voltage (Figure 7a) the electron transport before annealing could probably be improved by reducing the contact resistance and lowering the charge carrier trapping at dielectric/semiconductor interface.

To improve the transistor operation a self-assembled monolayer (SAM) treatment was applied in order to decrease the electron trapping at the semiconductor/dielectric interface. Figure 8 presents output characteristics of OFETs in BGTC configuration fabricated on the Si/SiO_2_ substrate modified by octadecyltrichlorosilane (OTS). The same switching behavior from electron to hole transport upon annealing was observed. The OTS monolayer improved the electron transport as evident from the maximum drain current being almost two orders of magnitude higher than without surface modification. Moreover, an evident saturation region emerged in the output characteristics at lower source-drain voltages. The improved electron charge carrier mobility of 1.5 × 10^−2^ cm^2^/(Vs) was only one order of magnitude lower than for PDI8-CN_2_ single crystal devices [38]. At the same time, the annealed BHJ films on OTS showed a similar hole mobility of 1 × 10^−3^ cm^2^/(Vs) to devices without surface modification (Appendix A, Appendix A). The shape of the p-type output characteristics after annealing indicated additional contact resistance (Figure 8d). Reduced trapping and enhanced operation of the n-type semiconductor indicated problems with the hole injection into PBTTT-C_14_ [18]. The lower hole mobility in the blend films in comparison to literature values for plain PBTTT-C_14_ [35] was related to the presence of PDIC8-CN_2_ decreasing the polymer order as proven by GIWAXS (Figure 6).

## 4. Conclusions

BHJ films were solution-processed using a blend of two semiconductors, p-type polymer PBTTT-C_14_ and n-type small molecule PDI8-CN_2_ applied as an active layer in OFETs. The thin film morphology depended on rotation speed, solution temperature and thermal annealing. In the as-cast BHJ films, the semiconductors phase separated into a disordered PBTTT-C_14_ phase and a PDI8-CN_2_ crystal network leading to a unipolar electron transport in OFETs. During annealing, less than 25 µm long PDI8-CN_2_ crystals were grown on the top of the film surface. Since the length of the crystals was smaller than the channel length of the OFETs, the electron transport between source and drain electrodes was interrupted, while the hole conduction was established through the PBTTT-C_14_ polymer phase. By changing the processing parameters of the BHJ films, especially by the post-treatment, the transistor operation is switched from unipolar n-type to unipolar p-type. This concept can be further developed for a new generation of high performance and unipolar blend films for complementary circuits. Changing the unipolarity of OFETs after fabrication would simplify the fabrication of inverters on large-area substrates. Large-area substrates could be covered by the active BHJ film in one solution-processing step and then subsequently annealed leading to the array of p- and n-type devices.

## Figures and Tables

**Figure 1 polymers-12-02662-f001:**
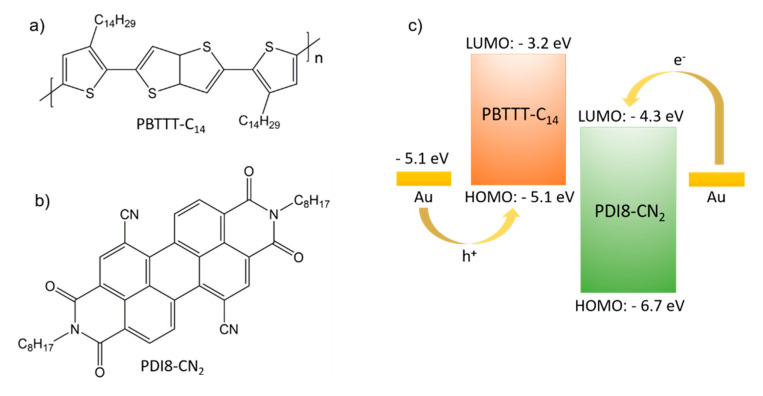
Chemical structures of (**a**) p-type PBTTT-C_14_, (**b**) n-type PDI8-CN_2_ and (**c**) an energy level diagram of the bulk heterojunction (BHJ) organic field-effect transistor (OFET).

**Figure 2 polymers-12-02662-f002:**
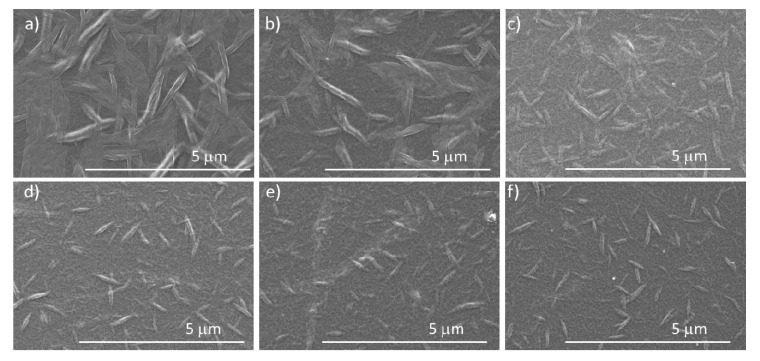
SEM images of PBTTT-C_14_:PDI8-CN_2_ BHJ films obtained by spin-coating from a 100 °C hot solution at rotation speeds of (**a**) 1000 rpm, (**b**) 2000 rpm, (**c**) 3000 rpm, (**d**) 4000 rpm, (**e**) 5000 rpm, (**f**) 6000 rpm.

**Figure 3 polymers-12-02662-f003:**
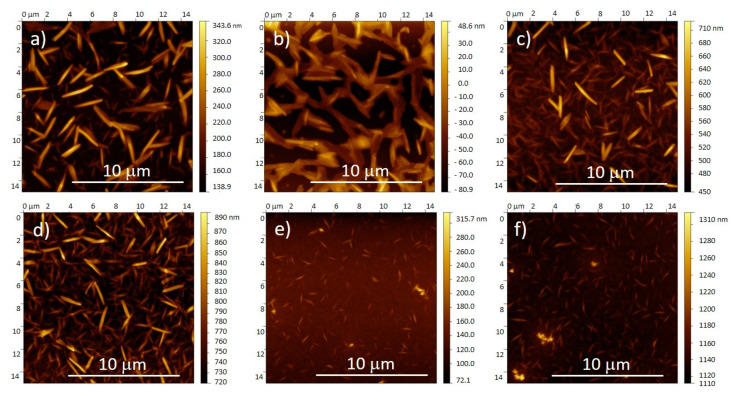
Atomic force microscopy (AFM) height images of PBTTT-C_14_:PDI8-CN_2_ BHJ films deposited at 3000 rpm and solution temperatures of (**a**) 80 °C, (**b**) 100 °C, (**c**) 120 °C, (**d**) 140 °C, (**e**) 160 °C, (**f**) 180 °C.

**Figure 4 polymers-12-02662-f004:**
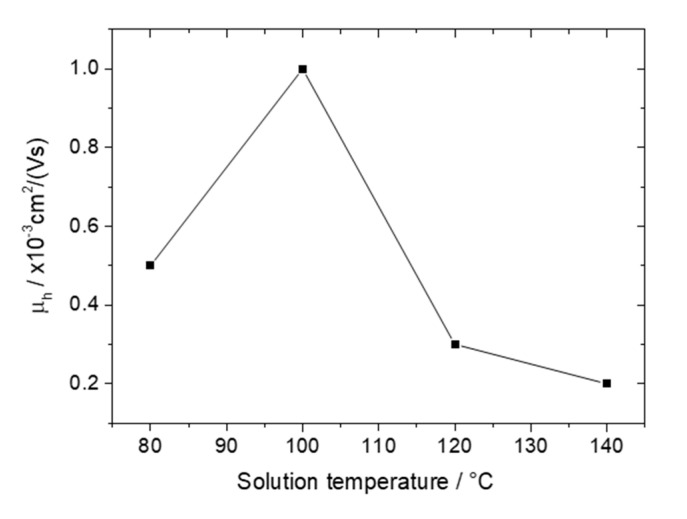
Field-effect mobility of holes as function of solution temperature for PBTTT-C_14_:PDI8-CN_2_ BHJ films deposited at 4000 rpm and thermally annealed at 140 °C.

**Figure 5 polymers-12-02662-f005:**
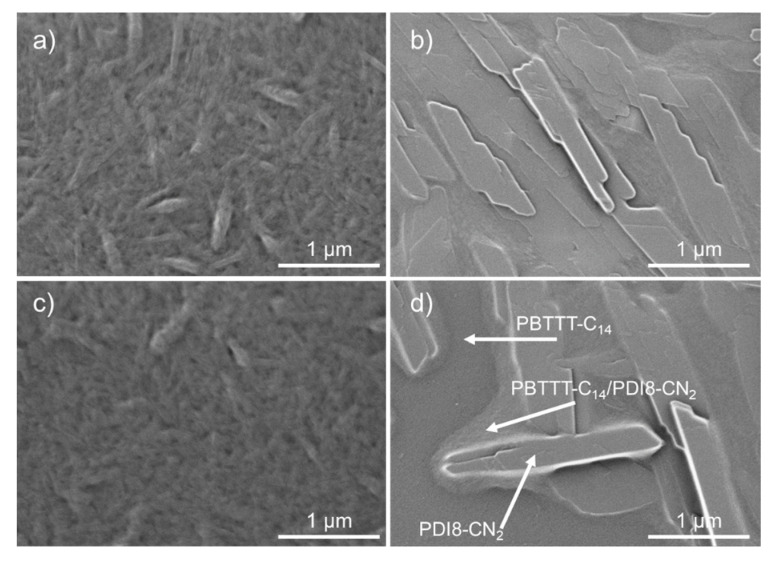
SEM images of PBTTT-C_14_:PDI8-CN_2_ BHJ films cast at solution temperatures of (**a**,**b**) 100 °C and (**c**,**d**) 120 °C; (**a**,**c**) as-cast, (**b**,**d**) after thermal annealing at 200 °C (the two phase separated fractions are indicated).

**Figure 6 polymers-12-02662-f006:**
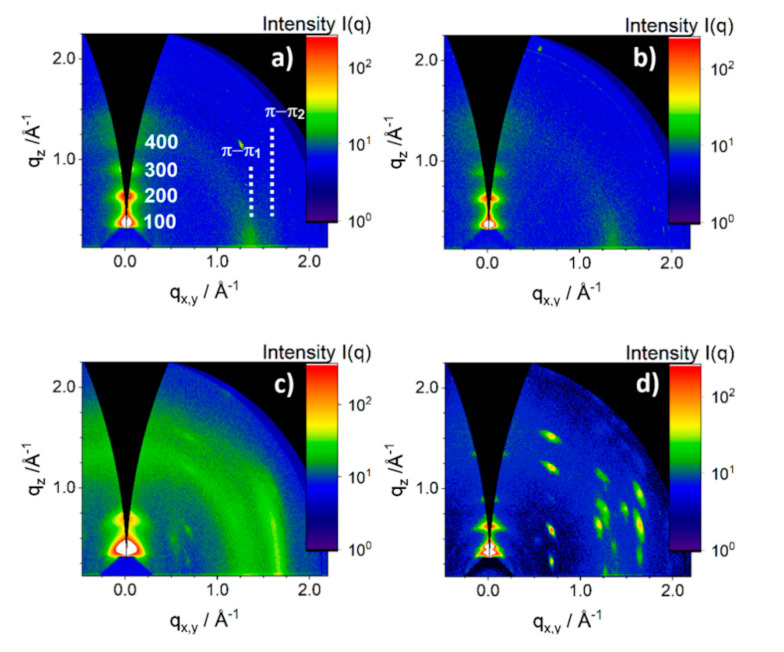
Grazing incidence wide-angle X-ray scattering (GIWAXS) patterns of PBTTT-C_14_ spin coated at 3000 rpm from a 100 °C hot solution (**a**) before and (**b**) after annealing at 200 °C and PBTTT-C_14_:PDI8-CN_2_ spin coated at 3000 rpm from solution at 100 °C (**c**) before and (**d**) after annealing at 200 °C. Numbers in the pattern (**a**) denote the peaks corresponding to h00 Bragg series of PBTTT-C_14_.

**Figure 7 polymers-12-02662-f007:**
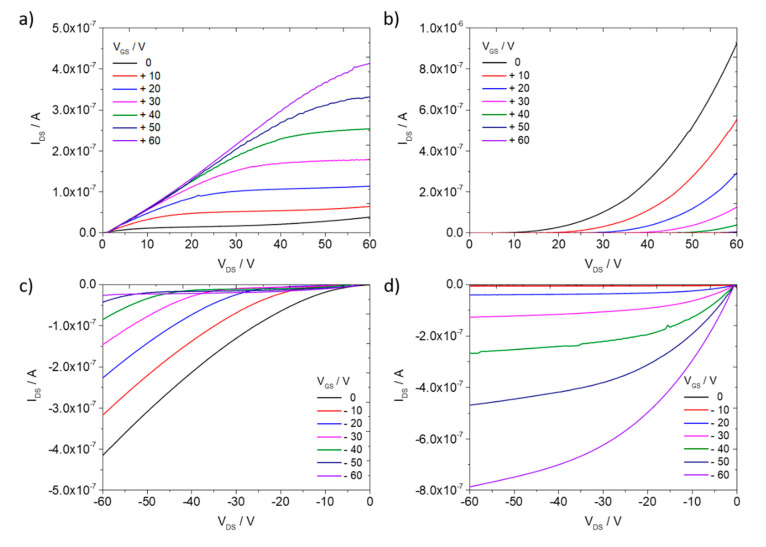
Output characteristics of OFETs based on PBTTT-C_14_:PDI8-CN_2_ BHJ films spin-coated at 4000 rpm and a solution temperature of 100 °C for (**a**,**b**) n- and (**c**,**d**) p-type regimes; (**a**,**c**) before and (**b**,**d**) after thermal annealing at 200 °C.

**Figure 8 polymers-12-02662-f008:**
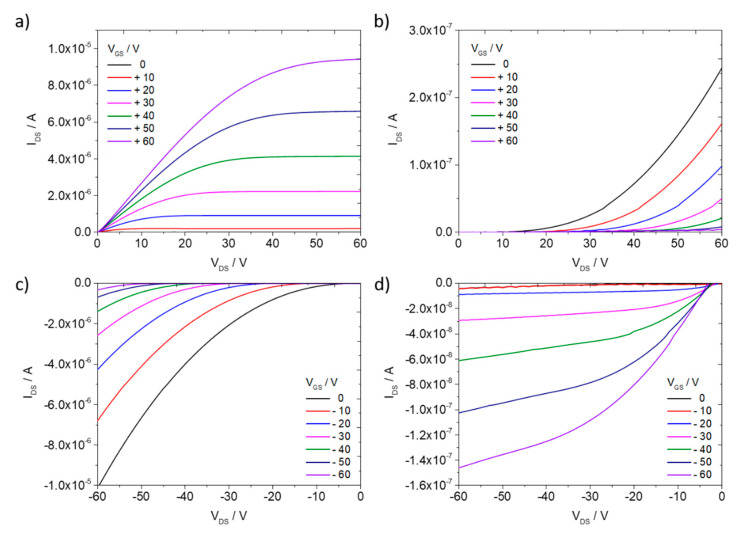
Output characteristics of OFETs based on PBTTT-C_14_:PDI8-CN_2_ BHJ film fabricated at 4000 rpm from 100 °C hot solution on OTS modified Si/SiO_2_ substrate for (**a**,**b**) n- and (**c**,**d**) p-type regimes; (**a**,**c**) before and (**b**,**d**) after annealing at 200 °C.

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
