# Peer review of "Switching from Electron to Hole Transport in Solution-Processed Organic Blend Field-Effect Transistors"

_polymers, 2020, doi:10.3390/polym12112662_

Round 1

Reviewer 1 Report

This manuscript entitled "Switching from electron to hole transport in solution-processed organic blend field-effect transistors'' presents a very important approach to switch from electron to hole transport in BHJ FETs using thermal post-treatment to control the occurred morphology changes. I recommend the manuscript for publication after major revision.

 while it provides new results in the field. However, I have some objections regarding the manuscript contents and therefore I recommend it to be published to Polymers, after major revision.

For the publication, the authors have to address my comments, as described below.

  • The English of the manuscript needs editing.
  • Novelty is not clearly pointed out.
  • PL is required to confirm charge transfer.
  • Which was the thickness of each formed thin film as a function of the rotation speed?
  • The effect of the film thickness and annealing in conductivity has to be investigated.
  • All figures are of low resolution and are not publishable. Please replace them with higher resolution ones.

Reviewer 2 Report

The authors demonstrate organic field-effect transistors based on the p-type polymer, PBTTT-C14, and the n-type small molecules, PDI8-CN2. They found that the electrical behaviors of the devices will tend to switch from n type to p type after thermal annealing. They also investigate the influence of solution and annealing temperatures on the OFET performance. Overall, this work may attract researchers in this field. However, the device mobility is relatively low, as compared to the values reported in the literature. There are also several issues need to be addressed before publication. Therefore, I cannot recommend to publish this work in the current state, but may reconsider after major revision. 

  1. The mobilities of PBTTT-C14 are two to three orders of magnitude lower than the results reported in the literatures (Heeney, M. et al. Nature Mater 5, 328–333 (2006). The authors should provide explanation for this large difference.
  2. This work spent much effort on the morphology controlling in the solution and thin-film states. It is suggested to provide more information for the solution status, ex. solution and thin-film UV-vis spectra or solution SAXS (if possible), to support their explanation. For the thin-film calculation, they should also provide DSC curves for these blends to investigate the crystalline morphology of the blends. 
  3.  The vertical phase separation may play an important role for the devices. The authors may investigate the vertical phase separation using SIMS (if accessible) or contact angles.  
  4. The authors mentioned that the device was not working above 140 degree. It is not reasonable because PBTTT should be thermal stable. The authors should provide more discussion for this. Also, it will be better if the authors use log scale in the y axis in Figure 4. It is hard to see the change in the low mobility value in Figure 4.
  5. The authors used GIXD to discuss polymer crystal domains. However, it is hard to see the difference from Figure 6d without one dimension integration curve. The authors should provide 1 D integration curve from X-ray and provide a table with detailed information (spacing and crystal size) of GIXD. 
  6. Most of figures (Figure 7 and 8) and images (Figure 6) show the low resolution. The authors should upload high-resolution figures for this work. 
  7. Thermally switching p-/n-type charge transport has been investigated in the literature (Y. Diao et al., J. Am. Chem. Soc. 2014, 136, 49, 17046). The authors should discuss the difference from their work and this paper.

Reviewer 3 Report

In this manuscript, the authors presented OFETs by BHJ composite thin film. The results reported in this manuscript are interesting, but BHJ thin film for OFETs was well studied. Thus the novelty of this manuscript is poor. The authors are strongly suggested to provide underlying physics of OFETs erformance versus different process conditions.

Reviewer 4 Report

The work shown in the manuscript "Switching from electron to hole transport in solution-processed organic blend field-effect transistors" deals with the transport properties of bulk heterofunction FETs obtained by blending an n-type and a p-type oligomer (PDI8-CN2 and PBTTT-C14, respectively).

The authors point out a correlation between the kind of charge transport (meaning hole channel vs. electron channel) and the morphology of the blended film, in particular with reference to the morphology of the uppermost layer of the film.

The topic is interesting in itself and the work is well enough presented. For these reasons, my overall evaluation if that the work is suitable for publication on Polymers in its present form.

Author Response

We would like to thank reviewer for his positive review.

Round 2

Reviewer 1 Report

The manuscript is very improved and my comments have been adequately addressed except from that for PL measurements, which is very significant towards the conclusions support. However, I recommend the publication of the manuscript.

Reviewer 2 Report

The authors have addressed the issues I mentioned last time. Therefore, this work could be considered for the publication as it is.